# Optimization of Ski Attitude for the In-Flight Aerodynamic Performance of Ski Jumping

**DOI:** 10.3390/biology11091362

**Published:** 2022-09-17

**Authors:** Lianzhong Cao, Youcai Guo, Xiong Li, Long Chen, Xin Wang, Tianyu Zhao

**Affiliations:** 1Department of Kinesiology, Shenyang Sport University, Shenyang 110102, China; 2Key Laboratory of Structural Dynamics of Liaoning Province, College of Sciences, Northeastern University, Shenyang 110819, China

**Keywords:** sports performance, biomechanics, computational fluid dynamics (CFD), optimization, kriging model

## Abstract

**Simple Summary:**

Flight distance determines the score of ski jumping and thus should be elongated in the competitions. Research has found that an analysis of aerodynamics is indispensable for ski jumping and the attitudes of athletes and skis are crucial factors to dictate the aerodynamic forces. As a follow-up to our previous computational fluid dynamics work, an optimization of the ski attitudes is conducted to maximize the lift-to-drag ratio under certain lift constraints. The optimal attitudes at these lift levels are of practical importance for athlete training and the angle of attack is proven to be pivotal for the optimal lift-to-drag ratio. The flow structures generated by the ski at the optimal attitudes are also discussed, together with a comparison with previous wind-tunnel measurements.

**Abstract:**

The control and adjustment of in-flight attitudes are critical to enlarging the flight distance of ski jumping. As one of the most important gears, the skis provide sufficient lift and drag forces for the athletes, and thus their in-flight attitudes should be optimized to improve flight performance. Here, the lift-to-drag ratio of a ski jumping ski is optimized with/without a constraint of lift capacity. The ski attitude is defined by three Eulerian angles and the resulting aerodynamic characteristics are predicted by Kriging models, which are established based on computational fluid dynamics (CFD) data. The surrogated models are dynamically updated in the optimization process to ensure their accuracy. Our results find that the optimization of the lift-to-drag ratio should be constrained by a certain lift capacity to be more practical. The angle of attack of the ski dominates the optimal lift-to-drag ratio at different lift levels while the yaw and roll angles are almost independent of the constraint once the required lift coefficient surpasses 0.6. This thus suggests that the athletes should focus on the angle of attack when modifying the ski attitude in the flight, which may reduce the difficulties in their in-flight decision makings.

## 1. Introduction

Ski jumping is one of the most competitive events in the Winter Olympic Games and is scored according to the jumping distance and athlete’s posture, in which the jumping distance is closely related to the aerodynamic performance of the body–skis system. Successful ski jumping is usually separated into four stages: in-run, take-off, in-flight, and landing [1]. First, the athlete transfers the potential energy into kinetic energy through in-run and reaches the initial speed of flight after take-off [2,3,4,5], followed by an adjustment towards the optimal flight posture within a short period. This optimal flight posture, including both body posture and ski attitude, is vital to sustaining excellent aerodynamic characteristics, thus maximizing the jumping distance. However, due to the complex degrees of freedom of the body–ski system, the coupling of unsteady aerodynamics and rigid body dynamics, and the effects of field gust, it is a challenging task to decide the optimal in-flight posture, especially ahead of field jumps.

For field jumps, it is laborious to measure the aerodynamic characteristics in the air within a short flight period. Thus, early studies conducted wind tunnel experiments as an alternative. Remizov et al. studied the optimal in-flight posture of ski jumpers based on wind tunnel measurement of aerodynamic forces [6]. It is shown that in the early flight, athletes should tilt their bodies forward and retain a low angle of attack (*α*) for the body to reduce drag. However, during the late flight, they should increase the *α* of the body to maximize the lift coefficient and thus obtain a longer jumping distance. Seo et al. measured the influence of the ski *α*, ankle angle, and the V-angle of the skis on aerodynamic force using wind tunnel experiments and therefore developed an aerodynamic model through data fitting [7]. They proposed that in the early flight, the main task of the athlete should be reducing the drag, while in the late flight, the focus should be replaced by maximizing the lift. They also found that the optimal V angle of the skis is about 26° to reduce drag in the early flight phase and to maximize lift during the rest of a jump. Later, as the V-style became dominant, scientists began to pay special attention to the effects of the V angle on ski aerodynamics, combined with the contribution of other attitude angles of ski, i.e., *α* and edge angle. Virmavirta and Kivekäs conducted wind tunnel measurements on the aerodynamic characteristics of an isolated ski [8]. Within a wide range of *α*, V-angle, and edge angle, the max lift–drag ratio of the ski is achieved around *α* = 30°, and a larger V angle will increase the sensitivity of ski aerodynamic forces to edge angle. In addition, they suggested that the optimal edge angle corresponding to the lift-to-drag ratio maximum at *α* = 30° should be limited to 5–10°. Despite that the research of Virmavirta and Kivekäs presented the detailed aerodynamic characteristics of an isolated ski, no flow field data were provided and thus less insight into the interplay of the aerodynamic forces and the flow structures has been uncovered.

Given the rapid development of computer science [9,10,11,12,13], computational fluid dynamics (CFD) simulation has become an important numerical tool in scientific research, the results of which are comparable to wind tunnel experiments. More importantly, CFD simulations can outline the flow structures, thus helping to understand the mechanisms in force generation. Meile et al. first compared the CFD results with wind tunnel measurements on the in-flight aerodynamic forces and gave a reasonable agreement [14]. Their comparison encouraged the application of CFD simulations in the aerodynamic analysis of ski jumping. Yamamoto et al. used the CFD method to simulate the influence of athlete posture on aerodynamic characteristics during take-off [15]. By analyzing the evaluation of vortex structure behind the athlete, they confirmed that the arm position has a remarkable impact on the aerodynamic performance. Kim et al. compared the performance of two flight postures via large-eddy simulation (LES) and separated the lift generation contributed from the body, legs, skis, arms, and head [16]. They also carried out parameterized research using their CFD-informed simplified model and found that optimal flight postures can increase the overall lift-to-drag ratio by 35% and 21% compared to the baseline postures. In the aerodynamic research of other sports, Defraeye et al. simulated the performance of different bicyclist postures using the CFD method, the accuracy of which was validated by wind tunnel experiments [17]. Barber et al. applied the CFD simulations to football aerodynamics and investigated the surface geometry effects on aerodynamic performance [18]. In swimming, Bixler et al. have conducted numerous research on the flow structures generated by the athlete. Additionally, the accuracy of CFD simulations in analyzing swimming hydrodynamics was confirmed [19,20]. All these studies support the usage of CFD simulation in analyzing the aerodynamic performance of athletes and their gears in various sports.

In addition, since a series of CFD simulations takes an extremely long time, which is common in engineering optimizations, the CFD simulations are often used to establish a surrogate aerodynamic model, which further combines with an optimization process to report an aerodynamically optimal posture (or gear structure) and thus assists the athlete training (or dear design). Taking the response surface approximation (RSA) model as the surrogate model, Shim et al. optimized the gaps between the bumpers and the ground and the leading angle of the front bumper in bobsleighs to minimize drag production [21]. Using the sequential quadratic programming (SQP) method, the drag coefficient of the optimal design was reduced by 3.08% compared to the reference design and the relative error is only 0.84% compared to a CFD validation. Gong and Gu combined the kriging model and a multi-island genetic algorithm to optimize the shape of the wind deflector of a tractor-trailer [22]. Their results showed that the optimized shape of the wind deflector can decrease the drag of the tractor-trailer by 4.65%, compared to the original design. The discrepancy between the CFD simulation and the surrogate model was 0.92% for the optimal design. Based on a CFD-informed Kriging model and the SQP method, Lee et al. optimized the ski *α* and the ankle angle of the body–ski system in ski jumping [23]. Compared to the reference model, the lift-to-drag ratio of the optimal posture was enhanced by 28.8%. Moreover, the accuracy of the optimization was validated via a further CFD simulation, and the error of the lift-to-drag ratio is 1.1%.

In our previous CFD research on ski attitudes [24], the relationship between the aerodynamic forces and the flow structures has been informed. As a follow-up, the current study employs our CFD method to generate a dynamic Kriging model, which further combines with a genetic algorithm to optimize the ski attitudes to maximize the lift-to-drag ratio under certain lift constraints. The numerical method and the optimization process are described in Section 2. The optimization result and flow analysis at the optimal ski attitude are provided in Section 3. The contribution of our results to the community and future directions are discussed in Section 4. Finally, our concluding remarks are summarized in Section 5.

## 2. Methods

### 2.1. Numerical Simulation

In this study, a simplified ski jumping ski is selected. As shown in Figure 1a, the ski (2.42 m × 0.11 m × 0.01 m in length, width, and thickness) consists of a rectangular flat plate and a semi-circle attached to the head, resulting in a reference area (*A*) of 0.2757 m^2^ [8]. The attitude angles of a ski have been defined in our previous research [24], and are labeled by *α* (equivalent to the angle of attack in the literature), *β* (half of the V-angle in the literature), and *γ* (equivalent to the edge angle in the literature). The ski is surrounded by a sphere fluid domain with a radius of 30 m (Figure 1b). All surface meshes are hexagons and the volume mesh is therefore generated via the polyhedral method with hex-cores. To better resolve the flow features near the ski and the aerodynamic loads on the ski surface, a concentric sphere region (radius = 4.25 m) with refined meshes is generated around the ski (Figure 1c). Two buffer layers are inserted between polyhedral and hexahedral volumetric meshes and the boundary layer constitutes 12 layers of hexahedral meshes, with the height of the first layer above the ski surface at 4 × 10^−5^ m (*y*^+^ ≈ 1).

The governing equations are the 3D implicit incompressible RANS equations,
(1)∇⋅v=0
and
(2)$$\frac{\partial v}{\partial t}+\left(v\cdot \nabla \right)v=f-\frac{1}{\rho }\nabla p+\upsilon \nabla^{2}v$$
where **v** is the velocity vector, **f** denotes the external force acting on the fluids, i.e., the gravity in this research. The air density (*ρ*) and kinetic viscosity (*ν*) are 1.2 kg/m^3^ and 1.5 × 10^−5^ m/s^2^, resulting in a Reynolds number (*Re*) at 4.8 × 10^6^. The governing equations are solved using the finite volume method. The solution of the momentum equation uses a second-order upwind scheme, and the pressure term adopts the second-order scheme. Under the condition that the result reaches the required accuracy, the temporal discretization is achieved by the first-order implicit formulation to improve the convergence speed. To account for the flow transition from laminar structures to turbulent structures, the *k*-*ω* SST (shear stress transient) model, as a combination of conventional *k*-*ε* and *k*-*ω* models [25], is introduced in the solution [23,26]. For boundary conditions, a uniform constant incoming airflow (*U*_∞_) with a magnitude of 30 m/s is imposed at the far-field of the fluid domain and the direction is set according to the ski attitude [24]. Moreover, the ski surfaces are all defined as non-slip walls. We further determine the most economic cell number of the computational mesh via a grid-dependence test (Figure 2a). Figure 2a shows the impact of cell number on the lift and drag coefficients, i.e., *C_L_* (*C_L_* = 2*L*/*ρU*_∞_^2^*A*, *L* is the lift) and *C_D_* (*C_L_* = 2*D*/*ρU*_∞_^2^*A*, *D* is the drag) of a ski at *α* = 30°, *β* = 15°, and *γ* = 0°, and the solution is almost converged when the total cell number reaches 1,000,000. Therefore, the computational mesh with 1,300,000 cells is used in this research.

### 2.2. Optimization Strategy

#### 2.2.1. Kriging Model

As an unbiased surrogate model, the Kriging model possesses excellent nonlinear data-fitting capability [27] and has been introduced into optimization works in many fields [28,29,30,31,32]. The Kriging model assumes the mapping between the objective function (*y*) and design variables (***x***) as a stochastic process:(3)$$y=f{\left(x\right)}^{T}k+z\left(x\right)$$

Here, *f*(***x***) denotes a regression model, emanating from a linear superposition of its sub-functions *f_i_*(***x***) (*i* = 1, 2, …, *p*, *p* is the dimension of design variables). The weight of each *f_i_*(***x***) is determined by *k_i_* (*i* = 1, 2, …, *p*). In addition, *z*(***x***) in Equation (3) introduces a random error to the regression. The mean value of *z*(***x***) is zero and the covariance:(4)$$Cov\left(z\left(x^{i}\right),z\left(x^{j}\right)\right)=\sigma^{2}R\left(\theta ,x^{i},x^{j}\right)$$
with
(5)$$R\left(\theta ,x^{i},x^{j}\right)=\prod_{l}^{p^{2}}R_{l}\left(\theta_{l},x_{l}^{i}-x_{l}^{j}\right)$$
*σ*^2^ is the process variance for each objective function *y*. *R* is the Gaussian correlation function, which is related to the weights *θ* and the distance between two sample points *x_l_^i^* − *x_l_^j^*. In this paper, the design variable ***x*** is defined as the attitude angles of the ski, ranging from 0° to 40° for each angle to cover all possible ski positions in an actual ski jumping, and the lift-to-drag ratio (*L*/*D*, also *C_L_/C_D_*) and lift coefficient are the objective functions. The sample points (125 in total) to establish the original Kriging models are uniformly distributed in the parameter space (Figure 2b) and the corresponding aerodynamic forces are pre-calculated via CFD simulations. During the optimization iteration, the Kriging model for *C_L_/C_D_* is dynamic, the accuracy of which is improved via adding two sample points during each optimization.

#### 2.2.2. Expectation Improvement (EI) Criteria

The precision of the Kriging model is critical to output an accurate estimation for optimization. Increasing the number of initial sample points is an efficient way to improve the overall accuracy of the Kriging model, but this often requires considerable computational resources for pre-calculation. An alternative is to establish a dynamic Kriging model that includes extra sample points into the original pool during each iteration of the optimization. In this research, two extra points are added in each iteration which is determined by the Expectation Improvement (EI) criterion and the transient optimal value [33].

The EI criterion is relevant to the global optimal value (taking a minimum problem as an example) and the uncertainty of the Kriging model,
(6)$$E\left[I\left(x\right)\right]=\left\{ \begin{array}{l} \left(y_{\min}-\hat{y}\left(x\right)\right)\Phi \left(\frac{y_{\min}-\hat{y}\left(x\right)}{\hat{s}\left(x\right)}\right)+\hat{s}\left(x\right)\phi \left(\frac{y_{\min}-\hat{y}\left(x\right)}{\hat{s}\left(x\right)}\right)\;\hat{s}\left(x\right)>0\\ 0\;\hat{s}\left(x\right)=0 \end{array}\right.$$

Here, Φ is the standard normal distribution, *φ* is the density of standard normal distribution, *y*_min_, and $\hat{s}\left(x\right)$ represent the global minimum and the root mean squared error (RMSE), respectively. Thus, the EI of each estimation from the Kriging model can be calculated. The point with the largest EI value is therefore selected as the add-in point for the current sampling pool. The actual aerodynamic forces for this point are then obtained via CFD simulations and then the Kriging model is updated according to the new sampling pool. The update of the Kriging model is terminated until the optimization for *C_L_/C_D_* converges, which is dictated by the residual of the EI criteria.

#### 2.2.3. Constraint of Lift Capacity

According to our previous aerodynamic analysis of an isolated ski [24], the *C_L_* of the ski becomes extremely low at a neutral *α* position, where the global peak of *C_L_/C_D_* is also observed. Therefore, the optimization of *C_L_/C_D_* for the ski attitude should be conducted under a constraint of *C_L_* minimum. Otherwise, the optimal attitude angles of the ski are not worthless since almost no lift capacity of the ski is retained. In our research, the penalty function method is used to impose the constraint of lift capacity. By setting a prescribed *C_L_* minimum, the objective function (*C_L_/C_D_*) that dissatisfies the constraint will be “penalized” to avoid its selection in the optimization. The constraint of *C_L_* is set as a series of values in our research to provide optimal ski attitudes at different *C_L_* levels.

#### 2.2.4. Optimization Process

The optimization procedure is illustrated in Figure 3. First, the design variables and the objective functions are selected. The original sampling pool is then generated and the corresponding values for the objective functions are calculated via CFD simulation. The kriging models for *C_L_/C_D_* and *C_L_* are thus constructed. Second, the genetic algorithm (GA) is employed to search for the optimal value of *C_L_/C_D_* (without a *C_L_* constraint), and the Kriging model for *C_L_/C_D_* is updated until convergence (the convergence threshold is set to *ε* = 0.001). The control parameters for the GA are as follows: the population is 100, and the rates for crossover and mutation are 0.3 and 0.08, respectively. To accelerate the convergence, we set 10° as the lower limit for *α* in the unconstrained optimization. Once the unconstrained optimization is completed, the *C_L_* constraint is applied to the optimized dynamic Kriging model for *C_L_/C_D_* using the penalty function method. The Kriging models for both *C_L_/C_D_* and *C_L_* are not updated in the constrained optimization since reasonable accuracy has been achieved.

## 3. Optimization Results

### 3.1. Unconstrained Optimization for C_L_/C_D_

In the unconstrained optimization for *C_L_/C_D_*, no *C_L_* constraint is imposed, and two new sample points are included in the pool according to the EI criteria and the optimal point (Figure 3). The optimization results are summarized in Table 1. Note that the iteration indicates the loop shown in Figure 3, instead of the optimization cycle in the GA. For the original Kriging model (iteration 0), the highest *C_L_/C_D_* found by the GA is 3.360 and the corresponding ski attitudes are 10.06°, 16.31°, and 0.3516°. A posterior validation for this optimal ski attitude via CFD simulation gives an error of around 0.1192%, which demonstrates the high accuracy of the current kriging model around the temporal optimal ski attitudes. Since the global optimum may deviate from this temporal point due to the error of the Kriging model in the parameter space, we calculate the EI criteria and search for the global EI maximum (0.07756), which is then simulated via CFD and included in the sampling pool to update the Kriging model. As the iteration marches, despite the abrupt increase in error (and EI maximum) at iterations 3 and 4, the Kriging model becomes more accurate within the entire parameter space since the EI maximum is continuously reduced until iteration 12 (Figure 4). The final Kriging model gives an EI maximum below *ε*. Meanwhile, the error of the Kriging model is mostly retained within 5% (except for iterations 3 and 4) throughout the update. The highest *C_L_/C_D_* estimated by the final Kriging model is 3.415 and the corresponding ski attitudes are *α =* 10.00° (the lower limit of *α*), *β* = 13.19°, and *γ =* 0.08791°. Compared to the original Kriging model, the optimal *C_L_/C_D_* is enlarged by 1.639%.

### 3.2. Constrained Optimization for C_L_/C_D_

The optimization results under different *C_L_* constraints are shown in Table 2. In our previous research [24], the *C_L_* maximum within the parameter space is around 0.82. Therefore, four *C_L_* levels below their maximum are selected consequently to demonstrate the effect of *C_L_* constraint on the optimization. In general, as the desired *C_L_* increases, the optimal *α* of the ski increases continuously, while the optimal *β* first boosts up to 31.62° at *C_L_* > 0.5 and then falls back to around 20° at higher *C_L_* constraints. The optimal *γ* also slightly increases as the *C_L_* constraint increases but finally stabilizes around 5° at *C_L_* > 0.7. It is thus indicated that the inclusion of the *C_L_* constraint mostly affects the optimal *α*. The optimal *C_L_/C_D_* also decreases at a higher *C_L_* constraint, which is mostly attributed to the concomitant enlargement of aerodynamic drag at a higher *α*. Again, the posterior CFD simulations validate the accuracy of the Kriging models for both *C_L_/C_D_* and *C_L_*. The errors of the optimal *C_L_/C_D_* under the highest three *C_L_* constraints are 0.06238%, 0.4345%, and 0.2403% (compared to CFD simulations) while the error in the optimization under *C_L_* > 0.5 is 3.787%. The errors of the optimal *C_L_* are 0.1126%, 0.6326%, 0.8256%, and 0.1595%, respectively.

### 3.3. Flow Field Analysis

The pressure distribution and flow structures of the ski at the optimal attitudes are shown in Figure 5. For the constraint of *C_L_* > 0.8 (Figure 5a), there are obvious low-pressure region (LRR) footprints on the dorsal surface of the ski, which deviates to the right edge due to the positive yaw and rolling angles. On the upwind side, the high-pressure region (HPR) is also asymmetric and deviates to the right edge. As the *C_L_* constraint decreases (Figure 5b–d), the LPR footprints on the dorsal surface are weakened due to the reduction in *α*, while the locations of these footprints are barely changed. Moreover, the LPR footprints near the aft edge of the ski experience a more remarkable attenuation. In addition, the variation of optimal *β* due to the decrease in *C_L_* constraint results in a trivial influence on the pressure distribution on the ski. Together with our previous simulations [24], it is thus inferred that the LPR footprints are formed when *β* achieves a threshold and a further change in *β* above the threshold cannot alter the footprint locations. When the constraint of *C_L_* drops to 0.5 (Figure 5d), most LPR footprints on the ski are not noticeable at the same strength level, except for the LPR at the ski head.

By using *Q*-criterion and *λ*_2_-criterion [34,35], the three-dimensional vortical structures produced by the skis at optimal attitude angles are shown in Figure 6 and Figure 7. In general, all optimal ski attitudes result in a tilted Hexa-vortex system or the coherent vortical structures above the dorsal surface of the ski, and the locations of these vortices are in good agreement with the LPR footprints (Figure 5). As the *C_L_* constraint decreases (Figure 6a–c), the strength of vortices beyond the mid-length is significantly attenuated. Moreover, the vortex #3 (V_3_) to #6 (V_6_) of the system are not resolved under the same level of dimensionless *Q* = 0.7 (Figure 6d). In contrast, vortex #1 (V_1_) is mostly retained at every *C_L_* constraint. Further, Figure 7 shows the attenuation of these vortices is accompanied by an elongation along the vortex line, leading to a slender vortex structure above the wing surface (Figure 7a–c). This can be related to the increased yaw angle *β* at a lower *C_L_* constraint, which further tilts away the vortex system (Figure 7d).

## 4. Discussion

Due to the high in-run speed and the transient variation of the incoming velocity during flight, it is not a simple task for ski jumpers to determine the optimal in-flight attitude angle for their skis. Predicting the optimal ski attitudes using the numerical method can not only reduce the training cost but also easily sweep over all possible statuses in the field jump, thus providing guiding significance for improving athletes’ flight skills.

In the aerodynamic research of ski jumping, the lift-to-drag ratio (*C_L_/C_D_*) is usually regarded as a critical index to evaluate the aerodynamic performance since the body–ski system undertakes a gliding flight after take-off. Under unconstrained conditions, the optimization reports an extremely high *C_L_/C_D_* for the ski. Under this attitude, the ski rotates to the lowest angle of attack (*α*) in our parameter space. Thus, the *C_D_* is significantly reduced as the pressure stress on the ski is significantly low and most of the drag comes from the friction stress. The low *C_D_*, therefore, leads to the high *C_L_/C_D_*. Moreover, since no obvious high-pressure region is generated on the upwind side of the ski, the limited pressure stress also contributes to a low *C_L_* (~0.2). Without a doubt, the most efficient way to enhance the aerodynamic performance of the ski is to achieve high *C_L_* and low *C_D_* simultaneously and thus leading to a high *C_L_/C_D_*. However, our analysis of the ski aerodynamics has shown that the *C_L_* and *C_D_* are mostly dependent on *α* and an increase in *C_L_* often triggers an increase in *C_D_* [24], which may finally lead to a shorter flight distance [36]. Usually, in the early flight phase, the athletes maintain a low *α* to reduce the *C_D_*. A low *C_D_* is particularly advantageous for retaining the initial flight velocity in the horizontal direction during the early in-flight period, which thus is an important factor in extending the flight distance [37]. During the later flight phase, the athletes focus on improving *C_L_*. A strong horizontal component of *C_L_* is beneficial for a larger flight distance [6,7]. In addition, although the horizontal component of *C_D_* is adverse to the flight, the vertical component of *C_D_* (against gravity) can support the athlete and extend the flight period [14]. This also explains why researchers often combine *C_L_* and *C_D_* (i.e., *C_L_/C_D_*) to assess ski aerodynamic performance. Considering the requirement of a high lift, therefore, the optimization of *C_L_/C_D_* for the ski should be conducted under the constraint of a minimum lift capacity. In this research, four different levels of *C_L_* constraints are selected according to the lift maximum of the ski and the search for optimal *C_L_/C_D_* is constrained under certain levels of lift capacity. It is found that *α* is the most important attitude angle to determine the lift and, in general, the optimal *C_L_/C_D_* is reduced as the *C_L_* constraint goes up. Moreover, to maintain the required lift capacity, the optimal *α* also increases for a higher *C_L_* constraint, while the optimal *β* first increases abruptly and then stabilizes around 20°. The optimal *γ* is suggested to be below 5° for all *C_L_* constraints, which agrees with the results of previous research [8], in which the *C_L_/C_D_* maximum is produced at *γ* = 5–10°.

Finally, the limitations of the current study are discussed. First, our sampling data originates from the numerical analysis of an isolated ski and ignores the interaction in a ski pair, as well as the interaction between the athlete’s body and skis. Second, our optimization of *C_L_/C_D_* only considers the constraint of lift capacity. In actual flights, the aerodynamic moments and the stability are also important to elongate the flight distance, thus being possible constraints for the optimization. Finally, the aerodynamic force of the ski is approximated by surrogate models. Despite that posterior CFD validations prove the accuracy of the surrogate models, fully CFD-based optimizations are still recommended given sufficient computational resources.

## 5. Conclusions

By combining the Kriging model and the genetic algorithm, the attitude angles of a ski jumping ski under conventional in-flight circumstances are optimized to enlarge the lift-to-drag ratio. The lift capacity of the ski can be constrained during optimization through the penalty function method. The Kriging models are established to reduce the computational resource and a dynamic upgrading strategy is employed to improve the accuracy. In the absence of lift constraints, due to the extremely low aerodynamic drag at a low angle of attack (*α*) around 10°, the ski can achieve a remarkably high lift-to-drag ratio. However, the corresponding lift generation is not high enough to enlarge the flight distance. When the lift constraints are imposed, *α* becomes one of the most critical attitude angles, in that the optimal lift-to-drag ratio at a higher lift constraint corresponds to a higher *α* but the yaw and roll angles become stabilized. The optimal yaw and roll angles are around 20°and 5°, respectively. This infers that the athletes can mostly retain the yaw and roll angles of the ski during flight while adjusting *α* to maintain the requirement of lift-to-drag ratio and lift capacity. This can provide further advice on the control of the skis for athletes, which may simplify their in-flight decision makings.

## Figures and Tables

**Figure 1 biology-11-01362-f001:**
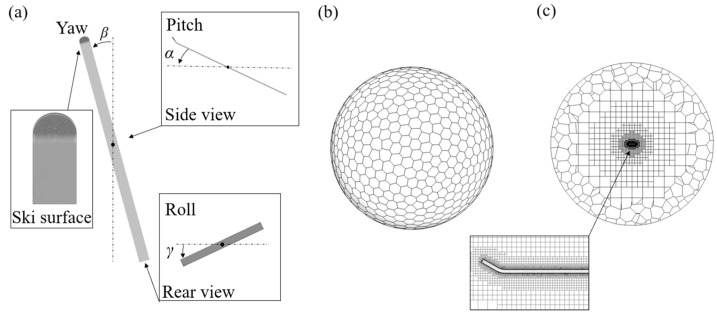
The ski attitude angles and computational mesh: (**a**) definition of attitude angles, (**b**) computational mesh of spherical fluid domain, and (**c**) mesh refinement around the ski.

**Figure 2 biology-11-01362-f002:**
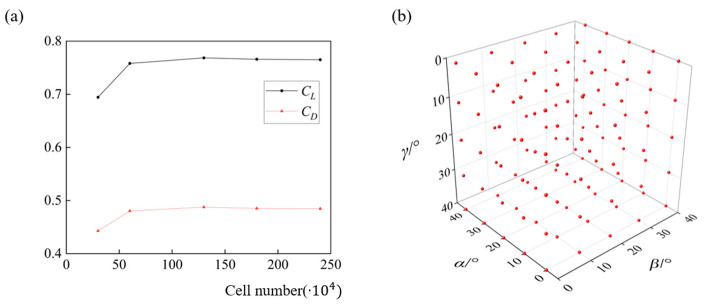
The mesh validation and sampling pool: (**a**) effects of cell number on the lift and drag coefficients of the ski and (**b**) distribution of 125 sample points to establish the Kriging models.

**Figure 3 biology-11-01362-f003:**
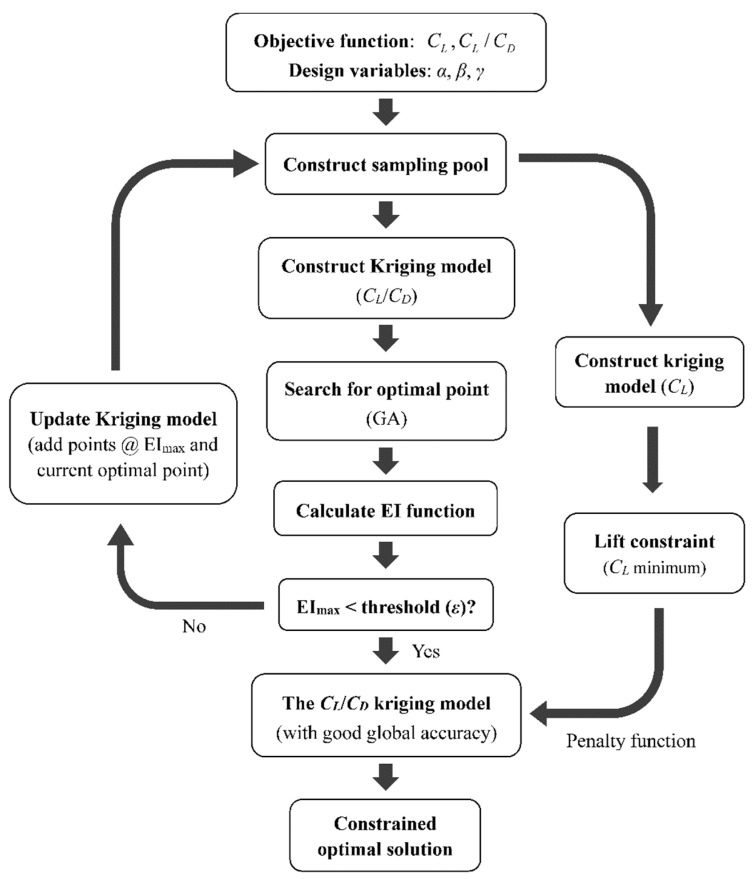
The optimization procedure of the current research.

**Figure 4 biology-11-01362-f004:**
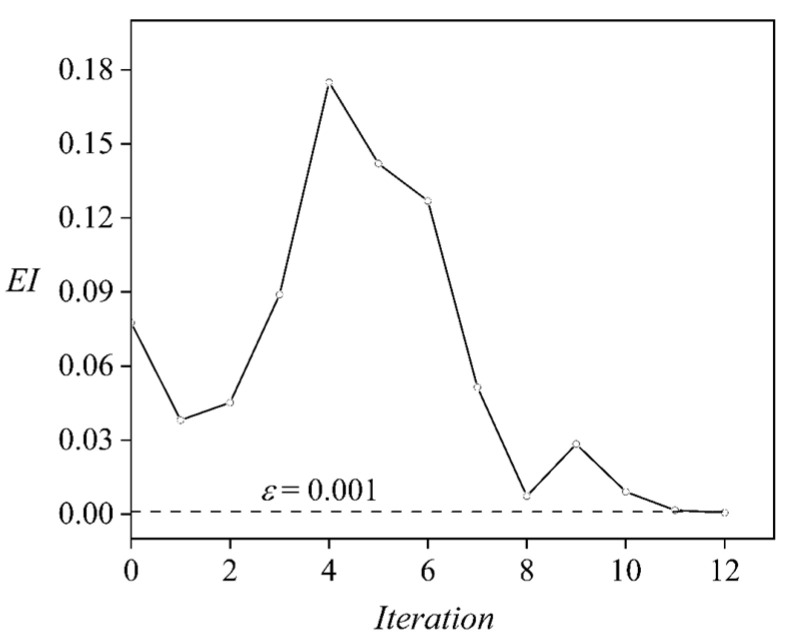
The EI Maximum of the dynamic Kriging model during the iterations. The dotted line indicates the EI threshold for convergence.

**Figure 5 biology-11-01362-f005:**
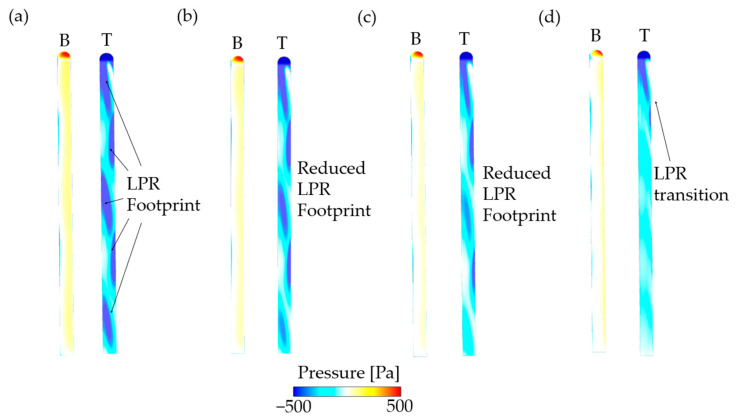
Pressure distribution of the ski at optimal attitudes: (**a**) *C_L_* > 0.8, (**b**) *C_L_* > 0.7, (**c**) *C_L_* > 0.6, (**d**) *C_L_* > 0.5. B: bottom view; T: top view.

**Figure 6 biology-11-01362-f006:**
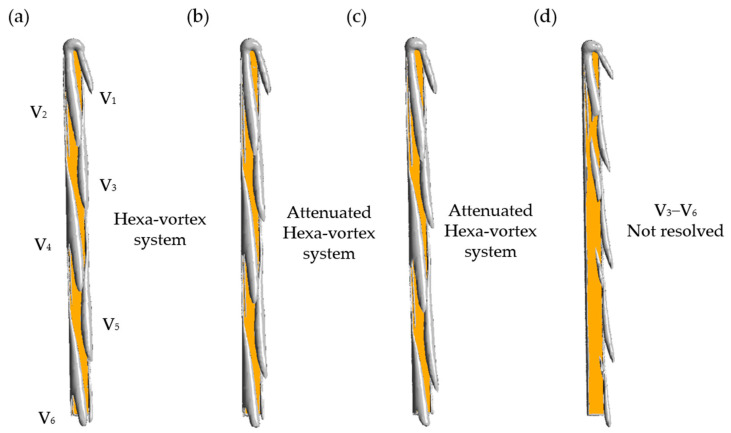
Three-dimensional vortex structure under different *C_L_* constraints: (**a**) *C_L_* > 0.8, (**b**) *C_L_* > 0.7, (**c**) *C_L_* > 0.6, (**d**) *C_L_* > 0.5. V: vortex. The iso-surfaces are outlined by dimensionless *Q* = 0.7. The ski is labeled by the yellow region.

**Figure 7 biology-11-01362-f007:**
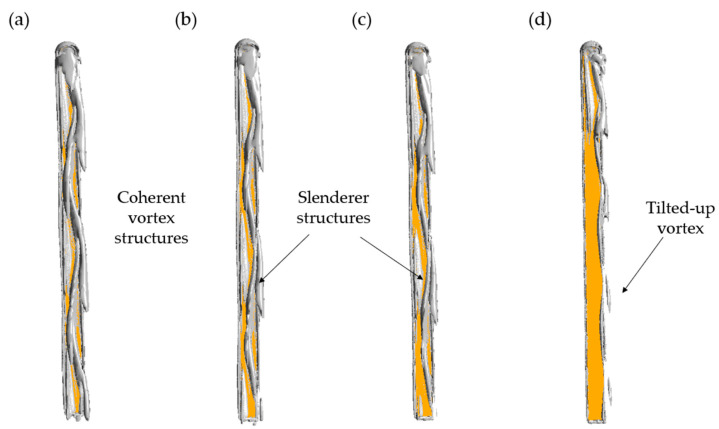
Three-dimensional vortex structure under different *C_L_* constraints: (**a**) *C_L_* > 0.8, (**b**) *C_L_* > 0.7, (**c**) *C_L_* > 0.6, (**d**) *C_L_* > 0.5. The iso-surfaces are outlined by dimensionless *λ*_2_ = −0.7. The ski is labeled by the yellow region.

**Table 1 biology-11-01362-t001:** Results of the unconstrained optimization. The iteration indicates the update of the dynamic Kriging model.

Iteration	*α*/°	*β*/°	*γ*/°	(*C_L_/C_D_*)*_KRG_*	(*C_L_/C_D_*)*_CFD_*	Error	EI_MAX_
0	10.06	16.31	0.3516	3.360	3.356	0.1192%	0.07756
1	10.00	17.39	0.09768	3.358	3.340	0.5389%	0.03810
2	10.54	23.70	0.08791	3.441	3.246	6.007%	0.04523
3	11.21	7.717	2.647	3.647	3.004	21.40%	0.08894
4	12.97	17.41	0.01954	3.604	3.020	19.34%	0.1750
5	10.00	15.89	2.803	3.481	3.301	5.453%	0.1421
6	10.43	13.38	0.2930	3.415	3.371	1.305%	0.1269
7	10.00	19.86	1.260	3.380	3.283	2.955%	0.05140
8	10.14	11.75	1.846	3.407	3.313	2.837%	0.007313
9	10.27	13.71	0.8107	3.379	3.370	0.2837%	0.02844
10	10.05	26.81	0.06837	3.484	3.297	5.672%	0.009051
11	10.03	13.66	0.4300	3.409	3.407	0.05870%	0.001609
12	10.00	13.19	0.08791	3.415	3.411	0.1173%	0.0005491

**Table 2 biology-11-01362-t002:** Results of constrained optimizations under different *C_L_* constraints.

Constraint	*α*/°	*β*/°	*γ*/°	(*C_L_/C_D_*)*_KRG_*	(*C_L_/C_D_*)*_CFD_*	Δ	(*C_L_*)*_KRG_*	(*C_L_*)*_CFD_*
*C_L_* > 0.8	27.58	21.69	5.695	1.604	1.603	0.06238%	0.8001	0.7992
*C_L_* > 0.7	23.68	21.81	6.251	1.849	1.841	0.4345%	0.7001	0.6957
*C_L_* > 0.6	21.68	20.67	3.370	2.086	2.081	0.2403%	0.6006	0.6056
*C_L_* > 0.5	16.21	31.62	1.631	2.494	2.403	3.787%	0.5007	0.5015
None	10.00	13.19	0.08791	3.415	3.411	0.1173%	0.1909	0.1921

## Data Availability

The data presented in this study are available on reasonable request from the corresponding author.

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
