# Peer review of "Optimization of Ski Attitude for the In-Flight Aerodynamic Performance of Ski Jumping"

_biology, 2022, doi:10.3390/biology11091362_

Round 1

Reviewer 1 Report

The peer-reviewed manuscript addresses the issue of ski length in the context of ski jumping performance. The term ski althitude is incorrect. The correct term is ski length. I do not understand the very idea of the paper. The authors omit from the construction of the research design the limitations of the athlete's ability to use skis. Ski length is strictly determined by the height and weight of the athlete.

Specifications for Competition Equipment Edition 2018/2019

1.2.1.1 Ski length
According to the body weight/height relationship (BMI = body weight / body height2 in kg/m-2,). See enclosure.
However, the maximum ski length is 145 % of the total body height of the competitor based on a minimum BMI of 21 for Ladies and 21 for Men. For athletes with less than minimum BMI a grading table of 0,125 BMI will be applied.
Exception: For Youth competitions, the maximum ski length is limited to 140 % from the body height only (no BMI formula will be applied).

The research in the manuscript is entirely theoretical in nature. The actual conditioning of the length of skis used by the athlete is omitted.  The design of the experiment itself and the analysis of the measured values is carried out on the basis of biomechanical evaluation. Although it can be assumed that the research work was carried out correctly, the result is useless. The experiment is strictly theoretical and I do not find any practical application of its results. I believe that the work is devoid of scientific value because it does not in any way add to the knowledge of ski jumping.

Author Response

 Response to Reviewer

We wish to thank the reviewer for providing detailed and insightful comments, and we hope all issues have been addressed satisfactorily.

  • Original comments are colored blue.
  • A point-to-point response can be found below.

The peer-reviewed manuscript addresses the issue of ski length in the context of ski jumping performance. The term ski althitude is incorrect. The correct term is ski length. I do not understand the very idea of the paper. The authors omit from the construction of the research design the limitations of the athlete's ability to use skis. Ski length is strictly determined by the height and weight of the athlete.

The research in the manuscript is entirely theoretical in nature. The actual conditioning of the length of skis used by the athlete is omitted.  The design of the experiment itself and the analysis of the measured values is carried out on the basis of biomechanical evaluation. Although it can be assumed that the research work was carried out correctly, the result is useless. The experiment is strictly theoretical and I do not find any practical application of its results. I believe that the work is devoid of scientific value because it does not in any way add to the knowledge of ski jumping.

  1. The explanation for the ‘ski length’.

Reply 1)

We agree with the reviewer that the ski length (or ski size) should be related to the height and weight of the athlete. In our research, the ski length is identical to a real ski jumping ski introduced in previous research Aerodynamics of an isolated ski jumping ski. (Mikko Virmavirta et al. 2019) and no further variation on the length has been applied.

  1. An explanation for the ‘ski attitude’.

Reply 2)

We suspect that the reviewer obfuscates the “ski attitude” with the “ski altitude”. In our research, the ski attitude refers to the position angles of the ski (α, β, γ) which determines its attitude concerning the global frame and the incoming flow. We do not include “ski altitude” in the current research.

  1. Justification for the research background and perspectives.

Reply 3)

For field jumps, athletes usually get a longer flight distance by adjusting the attitude angle of their bodies and skis and thereby improving aerodynamic performance. Since it is not easy to acquire the aerodynamic characteristics and flight data during actual field jumps, CFD simulations have gradually become an important alternative tool to explore aerodynamic characteristics. At present, CFD simulations have been widely introduced into sports research, such as ski jumping, swimming, football, etc. (Aerodynamic Analysis on Postures of Ski Jumpers during Flight using Computational Fluid Dynamics. Minhyoung RYU et al. 2015; Aerodynamics of ski jumping: experiments and CFD simulations. W. Meile et al. 2006). Further biomechanical and biomechanics analysis of these sports can be carried out on this basis.

Therefore, we employ a simplified ski model to carry out CFD simulation and establish the aerodynamic database for an isolated ski. In addition, to reduce the computing resources when exploring a large number of ski attitudes, we construct a surrogate model based on a limited number of aerodynamic data points and then predict the optimal ski attitude to enlarge the lift-to-drag ratio. We are confident with our results in this research, and believe it can provide significance for posture adjustment during actual ski jumping.

We are sorry for all the confusion led by the previous manuscript and hope this response can explain our motivation and contributions to ski jumping.

Reviewer 2 Report

This reviewer has no significant complaints concerning the manuscript.  I do have one minor issue that should be addressed.

1.  In tables 1 and 2 the number of significant digit reported is not consistent.  These are reported values from CFD and there is no value to reporting numbers past 3 significant digits.  The tables should be adjusted accordingly.

Author Response

 Response to Reviewer

We wish to thank the reviewer for providing detailed and insightful comments, and we hope all issues have been addressed satisfactorily.

  • Original comments are colored blue.
  • A point-to-point response can be found below.
  • All major changes in the manuscript are highlighted.

The reviewer has no significant complaints concerning the manuscript. I do have one minor issue that should be addressed.

  1. In tables 1 and 2 the number of significant digit reported is not consistent. These are reported values from CFD and there is no value to reporting numbers past 3 significant digits. The tables should be adjusted accordingly.

Reply 1)

The number of significant digits is uniformly adjusted to 4 throughout the manuscript.

Reviewer 3 Report

The authors optimize the ski attitude for the in-flight aerodynamic performance of ski jumping. The aim of the work is clear, and the subject is interesting. To me, the paper might deserve publication after the authors have replied to the three following remarks:

1) Some numerical details are not provided. Which discretization scheme do you use for the pressure term in the momentum equation? Please justify also why the temporal discretization is achieved by the first order implicit formulation and not a second order one. 

2) Grid independence studies should be described. 

3) The authors use a non-dimensional Q criterion to visualize the vortical structures. Nevertheless, Figure 6 is not enough clear to me. I suggest using also the lambda2 criterion (Jeong and Hussain, 1995) to identify the coherent vortical structures. Indeed, lambda2 is a very powerful tool in the comparison between the different configurations in terms of vortical structures as done, e.g., in Mariotti et al. (2019). Authors should show the results obtained with both criteria or, at least, mention also the lambda2 criterion and include the related references. 

References: 

J. Jeong, F. Hussain, On the identification of a vortex, J. Fluid Mech. 285, 69–94 (1995)

A. Mariotti, C. Galletti, E. Brunazzi, and M.V. Salvetti, “Steady flow regimes and mixing performance in arrow-shaped micro-mixers,” Physical Review Fluids 4, 034201 (2019)

Author Response

 Response to Reviewer

We wish to thank the reviewer for providing detailed and insightful comments, and we hope all issues have been addressed satisfactorily.

  • Original comments are colored blue.
  • A point-to-point response can be found below.
  • All major changes in the manuscript are highlighted.

The authors optimize the ski attitude for the in-flight aerodynamic performance of ski jumping. The aim of the work is clear, and the subject is interesting. To me, the paper might deserve publication after the authors have replied to the three following remarks:

  1. Some numerical details are not provided. Which discretization scheme do you use for the pressure term in the momentum equation? Please justify also why the temporal discretization is achieved by the first order implicit formulation and not a second order one. 

Reply 1)

Some numerical details are added in the revised manuscript (see highlighted lines 148-152). The pressure term adopts a second-order scheme and the gradients are calculated using the Green-Gauss Cell Based method. The following figure compares the results of two temporal discretization methods, i.e., the first-order implicit formulation and the second-order implicit formulation. There is less difference between the two methods, and thus the first-order implicit formulation is selected to improve convergence speed.

Fig. S1 Impacts of temporal discretization method on the lift and drag coefficients.

  1. Grid independence studies should be described. 

Reply 2)

 The grid independence studies have been further introduced in detail (see highlighted lines 158-163).

  1. The authors use a non-dimensional Q criterion to visualize the vortical structures. Nevertheless, Figure 6 is not enough clear to me. I suggest using also the lambda2 criterion (Jeong and Hussain, 1995) to identify the coherent vortical structures. Indeed, lambda2 is a very powerful tool in the comparison between the different configurations in terms of vortical structures as done, e.g., in Mariotti et al. (2019). Authors should show the results obtained with both criteria or, at least, mention also the lambda2 criterion and include the related references.

Reply 3)

We have added the λ2-criterion to identify the coherent vortex structures in our research (see highlighted lines 305-305).

Round 2

Reviewer 1 Report

Because the work has not been corrected according to my suggestions I cannot give a positive opinion.

Reviewer 3 Report

accept in present form